# Relative Effectiveness of Cell-Cultured versus Egg-Based Seasonal Influenza Vaccines in Preventing Influenza-Related Outcomes in Subjects 18 Years Old or Older: A Systematic Review and Meta-Analysis

**DOI:** 10.3390/ijerph19020818

**Published:** 2022-01-12

**Authors:** Joan Puig-Barberà, Sonia Tamames-Gómez, Pedro Plans-Rubio, José María Eiros-Bouza

**Affiliations:** 1FISABIO, Área de Investigación en Vacunas, 46020 Valencia, Spain; 2Consejería de Sanidad, Dirección General de Salud Pública, Servicio de Información en Salud Pública, 47001 Valladolid, Spain; sonia.tamames@jcyl.es; 3Departament de Salut, Agència de Salut Pública de Catalunya, 08005 Barcelona, Spain; pedro.plans@gencat.cat; 4Hospital Universitario “Río Hortega”, 47001 Valladolid, Spain; eiros@med.uva.es

**Keywords:** adults, influenza, cell-cultured vaccine, egg-based vaccine, influenza vaccine, relative vaccine effectiveness, real word evidence, mutation, human/prevention & control, comparative study

## Abstract

Avian mutations in vaccine strains obtained from embryonated eggs could impair vaccine effectiveness. We performed a systematic review and meta-analysis of the adjusted relative vaccine effectiveness (arVE) of seed cell-cultured influenza vaccines (ccIV) compared to egg-based influenza vaccines (eIV) in preventing laboratory-confirmed influenza related outcomes (IRO) or IRO by clinical codes, in subjects 18 and over. We completed the literature search in January 2021; applied exclusion criteria, evaluated risk of bias of the evidence, and performed heterogeneity, publication bias, qualitative, quantitative and sensitivity analyses. All estimates were computed using a random approach. International Prospective Register of Systematic Reviews, CRD42021228290. We identified 12 publications that reported 26 adjusted arVE results. Five publications reported 13 laboratory confirmed arVE and seven reported 13 code-ascertained arVE. Nine publications with 22 results were at low risk of bias. Heterogeneity was explained by season. We found a significant 11% (8 to 14%) adjusted arVE favoring ccIV in preventing any IRO in the 2017–2018 influenza season. The arVE was 3% (−2% to 7%) in the 2018–2019 influenza season. We found moderate evidence of a significant advantage of the ccIV in preventing IRO, compared to eIV, in a well-matched A(H3N2) predominant season.

## 1. Introduction

Influenza is a severe yearly threat to human health [1], especially in those at increased risk due to age or underlying medical conditions [2]. Influenza A and B viruses are responsible for yearly seasonal epidemics. The economic and health impact of influenza epidemics depends on virus characteristics, population immunity and preventive measures [3]. It has been reported that A(H1N1)pdm09 has a severe effect on young adults [4], B/Victoria-lineage in children [5], B/Yamagata-lineage shows a bimodal age distribution, affecting older ages than B/Victoria-lineage [5], and A(H3N2) is especially severe in the elderly [6]. Each year, seasonal influenza epidemics cause worldwide an estimated 3 to 5 million severe illnesses and 290,000 to 650,000 deaths [7,8,9].

Vaccines are the main line of protection against influenza [10]. Still, vaccine effectiveness in preventing influenza-related illness ranges from mild to moderate, with 40% to 60% effectiveness due to virus variability [11], the mismatch between the vaccine and circulating strains, the primed immune response to similar previous influenza infections and the mild immunodeficiency associated to age or underlying conditions [12]. Avian mutations in vaccine strains obtained from embryonated eggs also impair vaccine performance [12,13,14].

In the last thirty years, in addition to the annual update of the vaccine composition to the presumed circulating strains [15], various strategies have been adopted to outdo the limitations of the traditional seasonal influenza vaccines, such as enhancing immunogenicity with new administration routes, mucosal or intradermal [16], the addition of adjuvants [17] or increasing the amount of antigen in the vaccine [18,19] and improving the vaccine strain match to circulating viruses by obtaining the vaccine antigen by recombination [20] or by culture in mammalian cells [21] to elude avian adaptative mutations along the vaccine manufacturing process [22].

Fully cell-cultured influenza vaccines (ccIV), from strain selection to manufacturing, were first licensed for their use in humans in the United States (U.S.) in 2016 [23]. For the influenza vaccines available before the 2019–2020 season, only the A(H3N2) component was obtained by cell-culture, while the A(H1N1)pdm09 and B vaccine components were obtained by egg-culture [24]. Cell-derived B lineages were added in the vaccines distributed in the 2018–2019 [25] and, since the 2019–2020 season, all the strains in the cell-culture vaccine used in the U.S. were obtained by cell-culture. The four strains cell-derived quadrivalent influenza vaccine was licensed in Europe for the 2017–2018 season [26].

A recent systematic review on the effectiveness of newer seasonal influenza vaccines [27] concluded that while it is assumed that ccIV may be more effective than traditional egg-based vaccines (eIV) due to reduced antigenic mutation during vaccine production, there are limited data to assess the effectiveness of ccIV compared with eIV and that the evidence regarding the comparability of newer vaccines to traditional seasonal influenza vaccines is uncertain due to a lack of available literature.

Various new recent publications, not included in the mentioned review, reported the adjusted relative vaccine effectiveness (arVE) of ccIV compared to eIV [28,29,30,31,32,33,34,35]. We performed a systematic review of the recently published evidence on relative vaccine effectiveness of ccIV compared to eIV in preventing influenza-related outcomes (IROs).

## 2. Materials and Methods

We followed the Preferred Reporting Items for Systematic Reviews and Meta-Analyses (PRISMA) guidelines for the protocol, conduct and reporting [36,37,38]. We registered the protocol in the International Prospective Register of Systematic Reviews (PROSPERO), CRD42021228290.

Our objective was to perform a systematic literature review of the existing evidence on the arVE of ccIV, compared to eIV in preventing A(H3N2) related IROs in subjects 18 years old or older. As secondary objectives, we looked to evaluate the arVE of ccIV, compared to eIV in preventing IRO with influenza, and IRO related either to A(H1N1)pdm09, B overall or B/lineage or IRO determined by specific clinical codes in subjects 18 years old or older. We defined IRO as any clinical outcome related to influenza, determined as IRO with laboratory-confirmed influenza or IRO with influenza-specific clinical codes.

We performed our search in PubMed, using the terms Influenza AND Vaccin* AND (Effectiv* OR Effic*) AND (relativ* OR compara*) AND cell, and subsequently adapted the search strategy to the Web of Science (WoS) All Databases and Core Collection, medRxiv and bioRxiv. In all cases, we performed the search since the inception of the databases and without restrictions in publication type or language. We aimed to saturation by checking the reference lists of relevant publications, scientific meetings, guidelines, reviews and the authors’ archives. Finally, we contacted the authors of retrieved publications for additional data, results or clarification. We repeated the same search strategy after data extraction, but before performing the data analysis.

### 2.1. Study Selection

We downloaded the search results into the Rayyan platform [39]. After trimming duplicates, two authors screened the titles and abstracts for inclusion. We excluded animal studies, case studies, immunogenicity studies, studies on pandemic or pre-pandemic vaccines, zoonotic vaccines, and considered for inclusion non-duplicated randomised clinical trials and non-randomised studies of the effect of interventions in humans, 18 years old and over; that reported arVE results in preventing IRO comparing ccIV with eIV. Each author judged independently and blinded to the other author’s selection the compliance with the inclusion criteria of the retrieved publications. After finishing this first selection, we unblinded the Rayyan platform. The four authors agreed on the publications for inclusion.

### 2.2. Data Extraction

From each publication that we considered for inclusion, we obtained the full text. We searched again in the full text according to criteria for duplicates, defined as publications with the same authors, season, population, age-groups and analysis. When we found duplicates, we chose papers in preference of conference proceedings or abstracts.

We extracted from the non-duplicated publications full text and supplementary documents the second name of the first author, journal or conference name, volume, first page or abstract number, publication year, season of reported results, geographical location, population source, age groups, study setting, study design; statistical method, measured outcomes, the method used to determine the outcomes (laboratory or clinical codes), control of confounders such as sex, race, underlying conditions, frailty, previous health care use, antivirals, days from symptoms onset to specimen collection, vaccination date, calendar time, previous vaccination, the method used to find out the vaccines administered, number of IRO and vaccinated subjects by vaccine type, arVE results, arVE confidence intervals (CI), and funding source. We checked again for compliance with inclusion criteria, excluded those results that did not comply with our inclusion criteria, and recorded the reasons for exclusion.

### 2.3. Study Quality Assessment

The four authors evaluated the quality of the publications independently by assessing the risk of bias (RoB) following the ROBINS-I guidelines and answering signaling questions using a modified ROBINS-I template [40].

We considered bias due to confounding, selection bias, classification bias, comparability and exchangeability, attrition bias, bias in the measurement of the outcomes, and outcome reporting bias. Each overall publication RoB was judged as low (the study was comparable to a well-performed randomised trial), moderate (the study was sound for a non-randomised study but could not be considered close to a well-performed randomised trial), serious (the study had some important problems) or critical (the study was too problematic to provide any valid evidence on the effects of the intervention). We resolved differences by consensus. We generated the plots showing the RoB by publication and domain using the ROBVIs tool [40]. We excluded publications with serious or critical RoB bias.

### 2.4. Data Analysis

We performed a descriptive analysis according to season, age group, study design, outcome setting (primary care, hospital or both), outcome determination method, funding and RoB.

Depending on the study design, the published results and their confidence intervals were incidence rates ratios, risk ratios or adjusted odds ratios. Under the rare-diseases assumption, we assumed these measures as unbiased estimators of the adjusted relative risk (aRR) [41], and arVE as (1 − aRR) × 100 [42], for the metanalysis calculations we entered the log of the aRR obtained from 1-(arVE/100), and the standard errors as (ln aRR upper limit − ln aRR lower limit)/3.92.

When we identified mutually-dependent results, and to avoid repeated contribution bias, we used the more encompassing and relevant arVE IRO result and report, in each case, the included and excluded results. We defined mutually-dependent results when a publication reported more than one arVE in preventing diverse defined IROS for the same age-group and season.

We evaluated the existence of outliers and heterogeneity among the included publications’ results by Galbraith and forest plots. After excluding mutually-dependent results, we considered the hypothesis of no difference in results between subgroups when the bilateral *p*-value for the Cochran’s Q statistic was >0.10 [43] and the presence of heterogeneity when I^2^ was ≥50% [44]. We explored the degree to which heterogeneity of the results was influenced by season, age, study setting, study design, IRO determination method, funding and RoB.

We explored publication bias and asymmetry of reported results with funnel plots and the test of Egger [45], considering all reported included results.

We performed a quantitative meta-analysis and computed the aggregate aRR, and aRR 95%CI when the following quality and homogeneity criteria were satisfied: low or moderate RoB, Q *p*-value > 0.10; I^2^ < 50%, no evidence of publication bias, and three or more results available after excluding mutually-dependent results.

We performed a sensitivity analysis of the quantitative meta-analysis by non-parametric trim and fill estimation of not included results and the imputed pooled result and repeated the same calculations without excluding mutually-dependent results [46]. We estimated the impact of not including one result, result by result, in the estimated pooled aRR and aRR 95%CI. Finally, we evaluated the effect of any outlier results on the overall results and their heterogeneity.

We performed all calculations under a conservative random-effects approach, assuming clinical and methodological heterogeneity between studies by restricted maximum likelihood [47] using STATA v. 17: StataCorp LLC. College Station, TX, USA.

## 3. Results

We performed the bibliographic search between the 7th and 12th of January 2021 and retrieved 5393 publications. We identified nine additional publications [48,49,50,51,52] in two recent influenza conferences (Options X 2019 and ESWI 2020) and the author’s datasets [30,53,54,55], adding to 5402 retrieved publications. Two authors screened the titles and abstracts, and the four authors reviewed and agreed on the screening result. We identified 18 publications [28,30,31,32,48,49,50,51,52,53,54,55,56,57] for full-text review (Figure 1). We performed a new search on the 1 April 2021 and did not identify additional publications.

Among the 18 publications, we excluded four conference abstracts [30,48,49,51] with duplicated results published in three journal papers [29,53,57]. The remaining 14 publications [28,29,31,32,33,34,35,50,52,53,54,55,56,57] reported 53 results (Figure 1, Appendix A). We checked compliance with reporting IRO in subjects ≥18 years. That process resulted in the exclusion of one publication [50] and a total of 18 results related to broadly defined outcomes or results on populations including paediatric age groups (Appendix A). We assessed the risk of bias in the remaining 13 publications, resulting in excluding one publication [31] and a total of nine results (Appendix A). Overall, the leading contributors to bias in the retrieved studies were the lack of information on the distribution of missing data, measurement of outcomes and selection of participants (Figure 2).

### 3.1. Qualitative Review of Included Publications

We included in our qualitative review 12 publications [28,29,32,33,34,35,52,53,54,55,56,57] that reported 26 results (Figure 1). We describe the main characteristics of the publications and the reported results in Table 1 and Table 2. All the included publications were conducted in the U.S. Seven publications reported 14 results in the 2017–2018 influenza season [28,29,32,34,35,54,55], four publications reported ten results in the 2018–2019 season [33,52,53,57], and one publication reported two results in the 2019–2020 season [56]. Eight publications reported 15 retrospective-cohort results [29,32,33,34,35,56,57], and four publications reported 11 test-negative results [28,52,54,55]. One publication reported three results in the 18 and over age-group [55]. Six publications reported seven results in the 18 to 64 age group [29,34,35,53,54,57], and seven publications reported 16 results in the ≥65 years of age-group [28,32,33,35,52,56,57]. Two publications reported three results in preventing primary care IRO [32,35], eight publications reported 18 results in preventing hospital IRO [28,29,32,33,52,53,55,56], and three publications reported five results in preventing IRO defined as medical encounters (encompassing outpatient and inpatient IROs) [34,54,57]. Five publications (one retrospective cohort and four test-negative) reported 13 results based on laboratory-confirmed IRO [28,34,52,54,55], and seven reported 13 results determined by clinical codes [32,33,34,35,53,56,57]. Five publications with eight results were pharma funded [28,29,35,53,57], whereas seven with 18 results were non-pharma funded [32,33,34,52,54,55,56]. Finally, according to RoB, nine at low RoB publications reported 22 results [28,32,33,34,52,53,55,56,57], and three at moderate risk of bias reported four results [29,35,58].

Two test-negative publications reported arVE results in preventing admissions with laboratory-confirmed A(H3N2) in the 2017–2018 season [28,55]. The point estimates of the two publications arVE results were non-significant, discordant, −4% and to 24.9%, and with overlapping intervals.

The previous two publications and two additional ones [28,34,54,55] also reported arVE for the 2017–2018 influenza season in preventing laboratory-confirmed outcomes with influenza overall, influenza A or B, the IROs were either admissions or medical encounters, one was in subjects aged 18 and over [55], two in subjects aged 18 to 64 [34,54] and one in subjects 65 and over [28]. Only one publication reported laboratory-confirmed results on arVE in preventing laboratory-confirmed outcomes in the 2018–2019 influenza season [52].

After accounting for multiple-dependent results, we report the pooled results estimates by IRO determination method and other potential confounders or effect modifiers in Table 2.

Overall the pooled aRRs ranged from 0.87 to 0.98 fovoring the cell-cultured vaccine (pooled arVE range of 2% to 13%). We observed substantial heterogeneity by season (*p* = 0.001). We could not reject the hypothesis of homogeneity in results by age-group (*p* = 0.340), study design (*p* = 0.740), outcome setting (*p* = 0.410), or funding (*p* = 0.550), and did not find evidence of differences between the pooled arVE results of laboratory-confirmed IRO compared to the pooled results obtained from code-ascertained IROs (*p* = 0.460) or risk of bias (*p* = 0.150).

### 3.2. Heterogeneity

We assessed heterogeneity by plotting the 26 retrieved results in a Galbraight plot (Figure 3). All results, except one (Boikos 2020b [35]), were inside the 95% confidence region.

We further assessed the heterogeneity with a forest-plot of the pooled results by season, age-group, outcome, study design, outcome determination method, funding and risk of bias (Figure 4). We restricted our analysis to not mutually-dependent results to avoid multiple contribution bias. We excluded results reported in only one study, such as those in subjects 18 and over [55] or the results for the 2019–2020 season [56]; finally, we also excluded the outlier result (Boikos 2020b [35]) identified in the Galbraith plot.

After applying the above restrictions, we could reject homogeneity by season, with a pooled aRR for the season 2017–2018 of 0.89 (95%CI 0.87 to 0.92), compared to 0.97 (0.93–1.02) for the 2018–2019 influenza season (Q, *p* = 0.00).

We observed significant results favoring the cell-cultured vaccine in the 2017–2018 season, in the 18 to 64 age group (0.93; 0.90 to 0.96), in preventing hospital-related IROs (0.92; 0.88–0.97), in the cohort-retrospective studies (0.95; 0.90–0.99), outcome derermination by code (0.94; 0.90–0.99), and studies at moderate risk of bias (0.90; 0.83–0.98).

With the same restrictions, we did not observe that other clinical or methodological factors such as adjustment for previous health care contacts, underlying conditions, frailty, previous vaccination, calendar time, and vaccination date or statistical method of data analysis had an impact on the homogeneity of the results (Appendix A).

### 3.3. Publication Bias

We did not detect evidence of publication bias or reporting asymmetry when considering the 26 included results (Egger test *p*-value of 0.9926; Appendix A). We performed the same statistical and graphical analysis by season, age group, study design, reported outcome, outcome determination method, funding and risk of bias and did not obtain evidence of plot asymmetry by the Egger test for any of the above factors, but some patterns emerged in the by factor funnel plots (Appendix A). We observed a homogenous distribution of retrieved results in the 2017–2018 and 2018–2019 seasons, but only two results in the 2019–2020 season [56]. We did not observe asymmetric distribution for the results in subjects ≥65, test-negative studies, influenza-related hospital outcomes, laboratory-confirmed results, non-pharma funded studies and results obtained in studies at low RoB. There was a lack of small studies results for the 18–64 age group; primary care and medical encounters (defined as contacts with either primary or hospital care) IRO; in the pharma funded studies, and the cohort-retrospective studies.

### 3.4. Meta-Analysis of the Published Relative Vaccine Effectiveness Results

After considering our predefined criteria for meta-analysis, we estimated the pooled arVE results for laboratory-confirmed IROs with influenza [28,34,54,55], (Figure 5a) and the pooled arVE regardless of the IRO determination method [28,29,32,34,35,54], in this instance, stratified by season and age group (Figure 5b,c).

#### 3.4.1. Adjusted Relative Vaccine Effectiveness in Preventing Laboratory-Confirmed IRO

When we pooled the results of the influenza laboratory-confirmed results obtained in the 2017–2018 season [28,34,54,55], we estimated a pooled aRR of 0.97 (IC95% of 0.86 to 1.09), with an overall heterogeneity (I^2^) of 0.0%, and homogenous results among publications (Cochran’s Q, *p* = 0.90); with a non-significant adjusted arVE of 3% (−9% to 14%) favoring the cell-cultured influenza vaccine compared to the egg-based influenza vaccine in preventing laboratory-confirmed IRO.

#### 3.4.2. Adjusted Relative Vaccine Effectiveness in Preventing Any IRO

The overall aRR in preventing any IRO in the 2017–2018 influenza season was 0.89 (0.87; 0.92) (Figure 5b), with an overall heterogeneity of 0.0%, with homogenous results across age groups (Cochran’s Q, *p* = 0.64). By age, the aRR was 0.91 (0.83–1.00) in the 18–64 age group (I^2^, 35.85 %), with no differences among results of the included publications, Q, *p* = 0.22, and 0.89 (0.86–0.92) in the ≥65 age group (I^2^ = 0.0%), with no differences among results of the included publications, Q, *p* = 0.58.

Overall, it translated to a significant arVE of 11% (8% to 13%) in preventing IRO favoring the cell-cultured versus the egg-based vaccine in the 2017–2018 season.

There were five results for the 2018–2019 season [33,52,53,57] with significant heterogeneity I^2^ = 79.10% and a Cochran’s Q *p* = 0.00 (Figure 5c). The overall result was an aRR of 0.97 (0.93–1.02). By age group, the aRR was 0.94 (0.92–0.95), I^2^ 0.02%, and Cochran’s Q, *p* = 0.68 in those aged 18–64, and 1.00 (0.96–1.05), I^2^ 40.92%, and Cochran’s Q, *p* = 0.28 in those aged 65 and over. Overall, the arVE of 3% (−2%, 7%), favoring the cell-culture vaccine with a significant result (6%; IC95% 5% to 8%) for the 18–64 age-group.

### 3.5. Sensitivity Analysis

In the non-parametric trim and fill analysis restricted to the 2017–2018 season, the predicted aRR interval with the observed and imputed studies was 0.89 (0.86–0.91), with three new imputed studies all favoring the cell-culture vaccine (Appendix A). When repeating the same analysis, including the excluded multiple dependent results, we obtained a similar effect with an estimated aRR of 0.90 (0.87–0.93) and two new imputed results (Appendix A). The trim and fill analysis for the 2018–2019 season resulted in one imputed non-retrieved small-size study favoring the cell-cultured vaccine with no differences in the reported and the imputed results (data not shown).

We estimated the impact of excluding result by result in the influenza seasons with enough results (2017–2018 and 2018–2019) and did not observe significant differences with the results contained in the estimated overall confidence interval for each season (Appendix A).

Finally, we studied the impact of the outlier result (Boikos 2020b [35]) in the estimates and heterogeneity of the 18–64 age group and overall results in the 2017–2018 influenza season. The aRR estimate was now of 0.88 (0.77, 1.00) for the 18–64 age group and 0.89 (0.86, 0.91) for the overall estimate, with an I^2^ of 70.46% and 0%, compared with the aRR of 0.91 (0.83–1.00) and 0.89 (0.87–0.92) and the heterogeneity (I^2^) of 35.85% and 0%, when we excluded this result as we show in the Figure 5b, resulting in a more precise estimate with lower heterogeneity for the 18–64 age-group, but with minor impact on the overall estimate.

## 4. Discussion

We identified 12 publications of non-randomised intervention studies that reported rVE of seed-cell ccIV compared to eIV in preventing IROs in the 2017–2018, 2018–2019 and 2019–2020 influenza seasons. The main source of heterogeneity on arVE estimates was the influenza season. We identified only three non-mutually dependent results on the arVE of ccIV compared to eIV in preventing A(H3N2) IRO, two in the 2017–2018 influenza season and one in the 2018–2019 influenza season. Following our secondary goal, we identified four publications that reported four homogenous laboratory-confirmed results obtained in different age groups in the 2017–2018 influenza season. The pooled arVE of these four results favored ccIV but was nonsignificant and with a broad confidence interval.

Finally, we identified six publications with six non mutually dependent, homogenous, with low heterogeneity, arVE results, three were real-world evidence studies, and three were laboratory-confirmed outcome studies. The pooled evidence was of the significant advantage of ccIV compared to eIV in preventing IRO in the 2017–2018 influenza season, regardless of IRO, outcome determination, setting or age group.

The results were null arVE for the 2018–2019 season, with substantial heterogeneity and strong evidence supporting age as a significant heterogeneity driver.

### 4.1. Interpretation and Validity

Overall our findings on homogeneity and heterogeneity by study characteristics are consistent with the existing evidence [58]. Others have explored and concluded the similarity of influenza vaccine effectiveness estimates obtained in primary care and inpatient settings, concluding that “no differences in VE estimates between inpatient and outpatient settings by studies using the test-negative design”. Regarding the age effect, a previous systematic review reported similar estimates by age group by type or subtype of influenza virus, with the variability by age in vaccine effectiveness estimates explained mainly by the different magnitude of influenza vaccine effectiveness by influenza virus type or subtype [11,12]. Finally, other authors have described the modifying effect of previous exposure by the birth cohort that may result in a negative interaction between vaccination with an unmatched strain and imprinted immunity [59]. This situation has been proposed to explain the age variability of estimated influenza vaccine effectiveness during the mixed 2018–2019 season [60].

While all the described evidence on homogeneity or heterogeneity has been obtained from tests-negative studies, we have additionally observed overall homogeneity in the arVE estimates obtained from low to moderate RoB tests-negative and retrospective-cohort RWE studies, an observation that has to be validated or rejected as more evidence accumulates from real-world data retrospective-cohorts studies in future influenza seasons.

We would expect the benefits of ccIV compared to eIV in seasons with limited antigenic drift and egg adaptation in the egg-derived vaccine strains [14,61]. This was the situation in the 2017–2018 season, A(H3N2) was the only cell-based strain in the ccIV vaccine, A(H3N2) was predominant, the vaccine strain was well matched with the circulating A(H3N2) strain, and egg adaptation occurred [62]. By contrast, the 2018–2019 influenza season was mixed, with A(H1N1)pdm09 accounting for 48% of subtyped strains and A(H3N2) for 49% [63]. Neither egg adaptation nor drift was observed for the A(H1N1)pdm09, but drift was observed in the A(H3N2), and similarity between circulating and vaccine strains was notably poorer for both the egg- and cell-based vaccine A(H3N2) viruses [64].

All in all, the mentioned influenza seasons characteristics are consistent with our meta-analysis results of a significant arVE favoring the ccIV in the 2017–2018 season and the no effect in the 2018–2019 season. Regarding the 2018–2019 season results, we observed overall significant heterogeneity and a non-homogenous effect by age. We must stress, nevertheless, that our overall pooled result was of no arVE in preventing IRO, in line with the fact that in the U.S. 2018–2019 influenza season, no advantage over the ccIV over the eIV was likely, as the A(H1N1)pdm09 strain in the 2018–2019 ccIV vaccine was egg-derived, and that the age-specific low estimates were reported in the U.S. for the 2018–2019 season in preventing A(H3N2), but not in preventing A(H1N1) pdm09 IROs [63], a fact that could explain the observed age non-homogenous results. The results in the 2018–2019 (and the only study in the 2019–2020) do reinforce the observation in the 2017–2018 season of a favourable impact, compared to the egg-based vaccine, of the seed-cell-based vaccine. When there is no mismatch between the vaccine and the circulating strains, egg adaptation has a negative impact. In the presence of a mismatch, it is the mismatch the main driver to low vaccine effectiveness.

### 4.2. Limitations

Caution is advisable in interpreting the meta-analysis of evidence from observational studies, given the risk of robust and precise but biased estimates [65]. We collected results for only three seasons, and in the 2019–2020 season, we identified only one study. This evidence availability will not change soon, as 2020, 2020–2021, 2021, South or North hemisphere influenza seasons have been absent, and the 2021–2022 influenza season is at the time of writing a question mark. In addition to the limited number of seasons, we must add the limitations of the methods in the included studies and our approach and restrictions to analyse the available evidence.

The laboratory-confirmed test-negative studies’ sources of bias reside in their small sample sizes and the limited virus subtyping. In addition to lack of precision and typing specificity, the poor information on how the subjects were enrolled is a critical point to judge the quality of test-negative studies, even when comparing vaccinated with vaccinated. In the absence of incidence density sampling or a well-run systematic recruitment process, critical selection bias cannot be discarded, jeopardizing the reliability of the retrieved test-negative studies, in which that information was usually poorly reported.

The main potential sources of bias in real-world evidence (RWE) studies are exposure determination, lack of information on confounders, outcome determination, proper adjustment and analysis. For the retrieved and recent RWE studies, exposure determination and information on confounders was adequate. The modelling and adjustment of real-world evidence studies have evolved significantly year by year. The use of propensity scores, inverse probability of treatment weighting, adjustment by calendar time and geographic region, and the use of Poisson regression provides a sound analytical framework. Nevertheless, we found a gap in reporting the number of subjects by group and the number of missing subjects by each of the analysed groups. In some of the studies, we missed calendar time adjustment and proper time-person analysis approaches. An additional weakness of RWE thus far is the specificity in the determination of IRO by clinical codes. Although, this should result in a non-differential classification bias and thus could only have an impact of a regression to the null on the arVE estimates. Supporting our argumentation and the overall results is, first, that it has been argued that when comparing observational study designs, imperfect specificity tends to under-estimate true vaccine effectiveness, but were similar across designs “except if fairly extreme inputs were used” [66], and, second, the good correlation between coding and actual influenza [67].

Regarding our approach, the judgments on RoB are, although well structured, qualitative and subject to researcher bias. In the homogeneity and heterogeneity analysis, our final attribution of the RoB category was not a significant effect modifier. We also decided not to include estimates in populations with high-risk conditions or by risk conditions. We argue that a focused, systematic review on special populations should be performed, and in our approach, the interest rested in the confounding and adjustment by underlying conditions. Moreover, analysis by high-risk conditions was reported only in few publications (Appendix A). We also decided to exclude mutually dependent results to avoid the overweighting of the results of the same exposure-population experiences on the results. The sensitivity and specificity of the statistical Q homogeneity test have been put into question, as in the presence of a small number of results, the Q test lacks the power to identify non-homogeneity, as could be the case for RoB (*p* = 0.15) due to the low number of results and studies classified of moderate RoB. In the presence of a large number of results, it offers false-positive results. And some authors counsel against it and prefer the I^2^ parameter [68]. We opted to report both as our database was in the middle of these two situations, but applying a conservative significance level alpha ≥0.1 instead of the conventional alpha ≥0.05 for the interpretation of the Q test to accept homogeneity.

### 4.3. Strengths

We followed the PRISMA guidelines for the study design, retrieval, selection, RoB analysis, reporting and analysis, with a detailed description of the included publications, homogeneity, heterogeneity, publication bias and sensitivity analysis, followed by a conservative random approach analysis. We performed the quantitative analysis only when homogenous, with low heterogeneity, and sufficient results were available. Accordingly, we did not pool results in the presence of heterogeneity, such as pooling of matched and mismatched seasons, as, in the current situation, the season was a clear modifying factor and pooling inadequate. In this scenario, the most relevant information is provided in the stratified by season analysis. We restricted our systematic review to the whole cell-derived vaccines to exclude the question of strain egg-adaptation. Although we included the absolute numbers of subjects by category in all but one of the included publications, we used only adjusted results. Here we agree with other authors that argue on the usefulness of unadjusted results [69].

## 5. Conclusions

Our systematic review provides low to moderate evidence supporting the ccIV advantage in preventing A(H3N2) related IROs compared to eIV in well-matched A(H3N2) predominant influenza season. Supports the use of well-powered real-world evidence studies to provide timely real-world evidence on the comparative effectiveness of different influenza vaccines in preventing relevant IRO. Mainly, we have collected evidence that in the presence of low risk of bias, the results are homogenous across settings, outcome determination methods, and study design.

## Figures and Tables

**Figure 1 ijerph-19-00818-f001:**
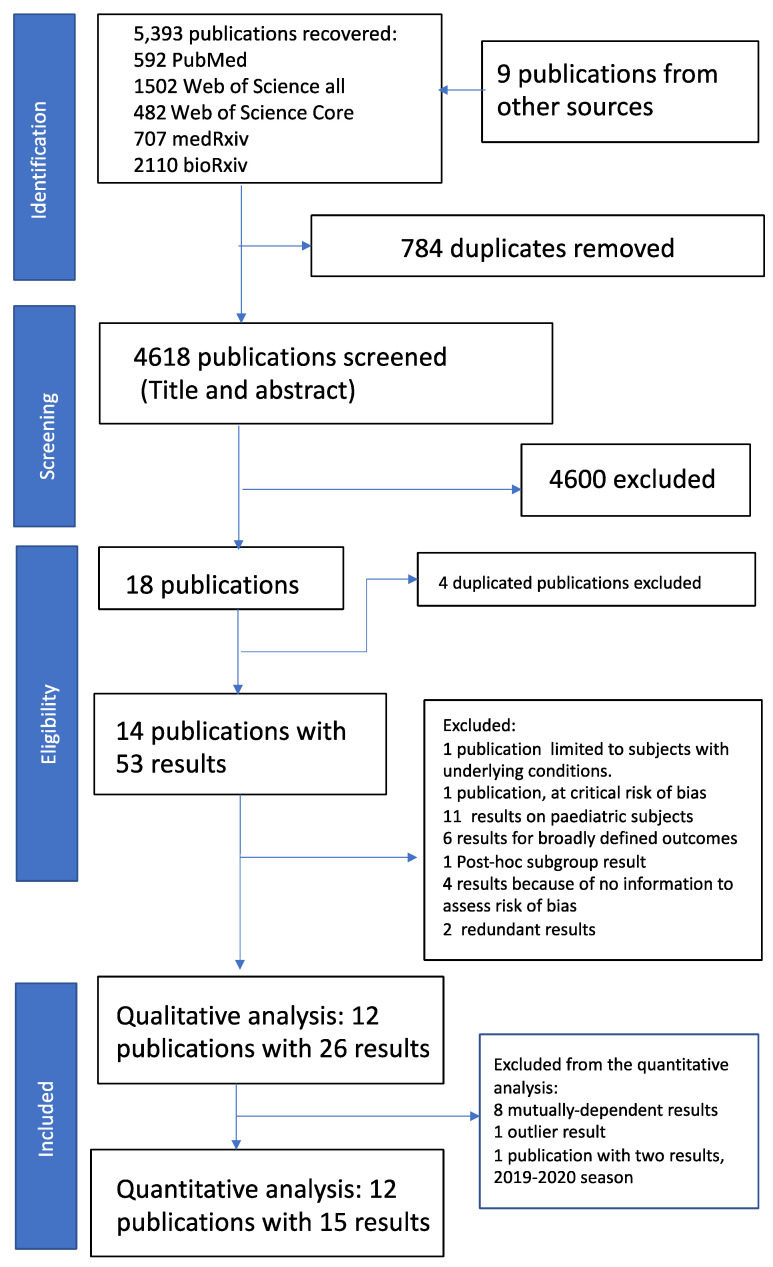
PRISMA flow diagram. See main text and Appendix A for details. PRISMA: Preferred Reporting Items for Systematic reviews and Meta-Analyses.

**Figure 2 ijerph-19-00818-f002:**
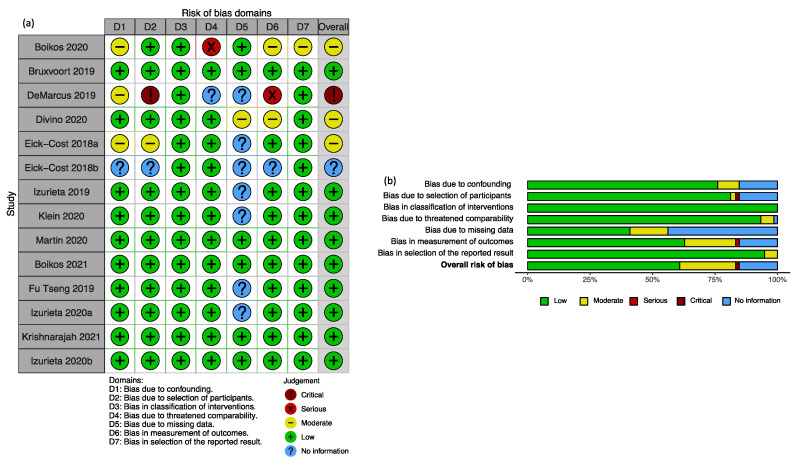
Risk of bias evaluation summary: (**a**) Judgement about risk of bias by domain in each recovered publication reporting results on the relative effectiveness of cell-cultured compared to egg-based vaccines. (**b**) Weighted contribution of each domain to the assessed risk of bias in the included publications.

**Figure 3 ijerph-19-00818-f003:**
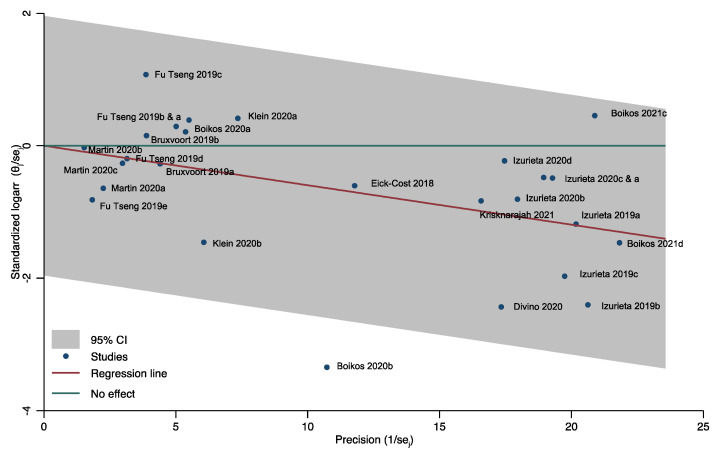
Heterogeneity and outliers among all included results. Relative vaccine effectiveness of cell-culture vs. egg-based vaccine in subjects ≥ 18.

**Figure 4 ijerph-19-00818-f004:**
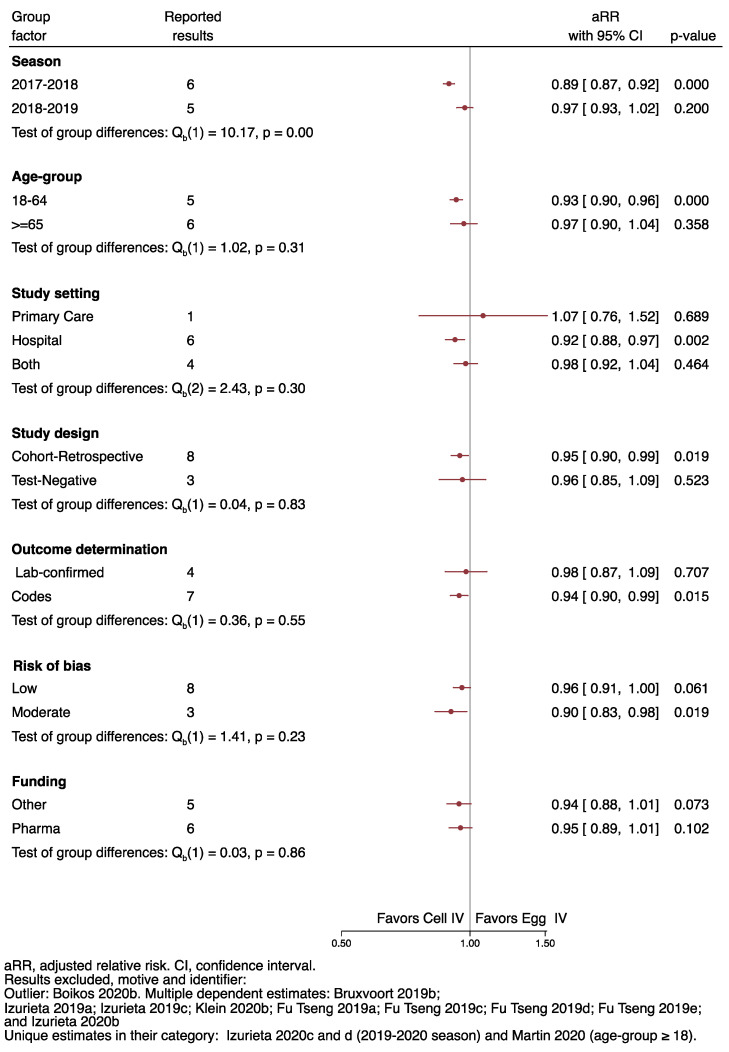
Heterogeneity of the results by season, age-group, study setting, study design, outcome determination method, risk of bias and funding.

**Figure 5 ijerph-19-00818-f005:**
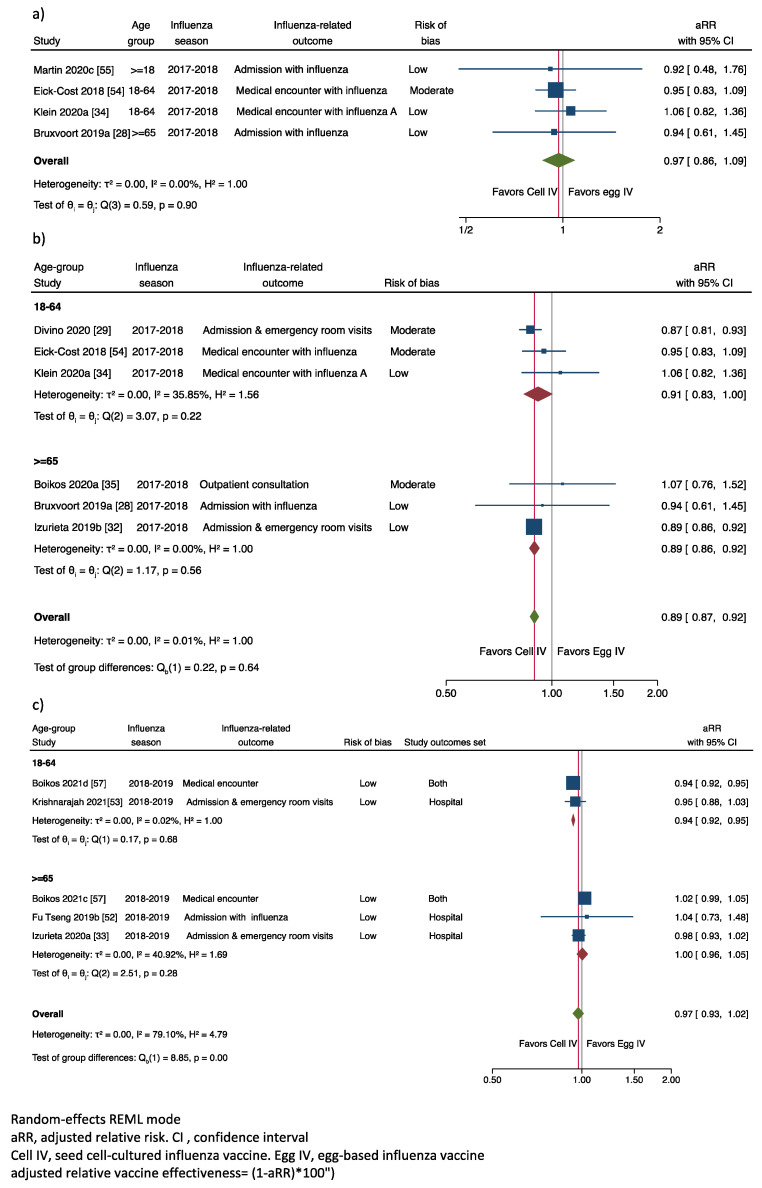
Adjusted relative risk of laboratory-confirmed influenza related outcomes comparing cell-cultured versus egg based influenza vaccines (**a**) Laboratory confirmed outcomes, 2017–2018 influenza season: (**b**) Code confirmed and laboratory confirmed outcomes, 2017–2018 influenza season; (**c**) Code confirmed and laboratory confirmed outcomes, 2018–2019 influenza season.

**Table 1 ijerph-19-00818-t001:** Studies included in the qualitative systematic review of adjusted relative vaccine effectiveness of cell versus egg-derived influenza vaccines in subjects 18 years old or older.

Author, Year	Season	Geographic Location	Study Design	Age Group	Outcome *	Outcome Determination Method ^†^	Risk of Bias	Cell-Cultured IV (n)	Influenza Related Outcomes (n)	Egg-Based IV (n)	Influenza Related Outcomes (n)	arVE (%)	arVE 95%CI		Funding
Boikos 2020a [35]	2017–2018	United States	Cohort-retrospective	≥65	Outpatient consultation	Codes	Moderate	29,618	521	164,151	4808	−7.3	−51.6	24	Seqirus
Boikos 2020b [35]	2017–2018	United States	Cohort-retrospective	18–64	Outpatient consultation	Codes	Moderate	55,104	1069	693,014	10,021	26.8	14.1	37.6	Seqirus
Bruxvoort 2019a [28]	2017–2018	United States	Test-negative	≥65	Admission with influenza	Lab-confirmed	Low	157	25	3498	612	6	−46	39	Seqirus
Bruxvoort 2019b [28]	2017–2018	United States	Test-negative	≥65	Admission with influenza, A(H3N2)	Lab-confirmed	Low	151	19	3321	435	−4	−70	37	Seqirus
Divino 2020 [29]	2017–2018	United States	Cohort-retrospective	18–64	Admission & emergency room visits	Codes	Moderate	499,156	976	1,730,403	4053	13.1	6.8	19	Seqirus
Eick-Cost 2018 [54]	2017–2018	Not reported	Test-negative	18–64	Medical encounter with influenza	Lab-confirmed	Moderate	2467	506	3239	757	5	−10	17	Defence Health Agency
Izurieta 2019a [32]	2017–2018	United States	Cohort-retrospective	≥65	Outpatient consultation	Codes	Low	659,249	3299	1,863,654	9607	5.7	1.9	9.4	FDA
Izurieta 2019b [32]	2017–2018	United States	Cohort-retrospective	≥65	Admission & emergency room visits	Codes	Low	659,249	4370	1,863,654	14,417	11	7.9	14	FDA
Izurieta 2019c [32]	2017–2018	United States	Cohort-retrospective	≥65	Admission	Codes	Low	659,249	2527	1,863,654	8359	9.5	5.3	13.4	FDA
Klein 2020a [34]	2017–2018	United States	Cohort-retrospective	18–64	Medical encounter, with influenza, A	Lab-confirmed	Low	40,685	.	712,126	.	−5.8	−36.1	17.7	DHHS
Klein 2020b [34]	2017–2018	United States	Cohort-retrospective	18–64	Medical encounter, with influenza, B	Lab-confirmed	Low	40,685	.	712,126	.	21.4	−7.3	42.4	DHHS
Martin 2020a [55]	2017–2018	United States	Test-negative	≥18	Admission with lab confirmed A(H3N2)	Lab-confirmed	Low	56	7	1459	248	24.9	−78.8	68.5	CDC, NIH
Martin 2020b [55]	2017–2018	United States	Test-negative	≥18	Admission with B/Yamagata-lineage	Lab-confirmed	Low	43	3	1135	83	1.8	−254	72.8	CDC, NIH
Martin 2020c [55]	2017–2018	United States	Test-negative	≥18	Admission with influenza	Lab-confirmed	Low	65	14	1676	399	8.5	−75.9	52.3	CDC, NIH
Boikos 2021c [57]	2018–2019	United States	Cohort-retrospective	≥65	Medical encounter	Codes	Low	517,639	6321	987,943	11,545	−2.2	−5.4	0.9	Seqirus
Boikos 2021d [57]	2018–2019	United States	Cohort-retrospective	18–64	Medical encounter	Codes	Low	1,529,189	24,084	5,384,922	87,113	6.5	5.2	7.9	Seqirus
Fu Tseng 2019a [52]	2018–2019	United States	Test-negative	≥65	Admission with influenza A	Lab-confirmed	Low	696	39	2773	146	−6	−54.3	28	Kaiser Permanente
Fu Tseng 2019b [52]	2018–2019	United States	Test-negative	≥65	Admission with influenza	Lab-confirmed	Low	696	39	2773	143	−4	−50.3	26	Kaiser Permanente
Fu Tseng 2019c [52]	2018–2019	United States	Test-negative	≥65	Admission with influenza A(H1N1)	Lab-confirmed	Low	696	22	2773	65	−32	−117	20	Kaiser Permanente
Fu Tseng 2019d [52]	2018–2019	United States	Test-negative	≥65	Admission with influenza A(H3N2)	Lab-confirmed	Low	696	13	2773	52	6	−75	49	Kaiser Permanente
Fu Tseng 2019e [52]	2018–2019	United States	Test-negative	≥65	Admission with influenza A untyped	Lab-confirmed	Low	696	4	2773	26	36	−86	78	Kaiser Permanente
Izurieta 2020a [33]	2018–2019	United States	Cohort-retrospective	≥65	Admission & emergency room visits	Codes	Low	761,037	2330	1,454,340	4582	2.5	−2.4	7.3	FDA
Izurieta 2020b [33]	2018–2019	United States	Cohort-retrospective	≥65	Admission	Codes	Low	761,037	1426	1,454,340	2790	4.4	−1.9	10.3	FDA
Krishnarajah 2021 [53]	2018–2019	United States	Cohort-retrospective	18–64	Admission & emergency room visits	Codes	Low	590,705	768	2,223,435	3113	4.9	−2.8	12.1	Seqirus
Izurieta 2020c [56]	2019–2020	United States	Cohort-retrospective	≥65	Admission & emergency room visits	Codes	Low	824,264	2092	1,584,451	3956	2.5	−2.8	7.6	FDA
Izurieta 2020d [56]	2019–2020	United States	Cohort-retrospective	≥65	Admission	Codes	Low	824,264	1255	1,584,451	2309	1.3	−5.7	7.9	FDA

* All outcomes are either influenza related or with laboratory confirmed influenza, see next column, outcome definition method. ^†^ Codes: International Classification of Diseases, Tenth Revision, Clinical Modification codes: J09 Influenza due to certain identified influenza virus. J10 Influenza due to other identified influenza virus. J10.0 Influenza with pneumonia, other influenza virus identified. J10.1 Influenza with other respiratory manifestations, other influenza virus identified. J10.8 Influenza with other manifestations, other influenza virus identified. J11 Influenza, virus not identified. J11.0 Influenza with pneumonia, virus not identified. J11.1 Influenza with other respiratory manifestations, virus not identified. J11.8 Influenza with other manifestations, virus not identified. On the three studies by Izurieta et al. the code J12.9 Viral pneumonia, unspecified was added to define the outcomes. Laboratory confirmed outcomes: all Real time polymerase chain reactions. IV, influenza vaccine. arVE, adjusted relative vaccine effectiveness. CI confidence interval.

**Table 2 ijerph-19-00818-t002:** Number of publications, results, mean number of subjects vaccinated, IRO cases, adjusted relative risk estimates, heterogeneity and test group differences after excluding mutually-dependent results *, by category of confounders or effect modifiers.

Confounders, Effect Modifiers	Publications Included		Reported Results		Cell-Cultured IV	IRO	Egg-Based IV	IRO	aRR ^¶^	(95% CI)		Heterogeneity	Test of Group Differences	
	n = 12 ^†^	%	n = 26	%	Mean	Mean	Mean	Mean				I^2^ (%) ^§^	Q **	*p*-Value
Season													16.86	<0.001
2017–2018	7	58.3	14	53.8	188,995	1111	686,936	4483	0.89	0.86	0.91	0.01		
2018–2019	4	33.3	10	38.5	416,309	3505	1,151,885	10,958	0.97	0.93	1.02	79.15		
2019–2010	1	8.3	2	7.7	824,264	1674	1,584,451	3133	0.98	0.92	1.02	0.00		
Age group													2.16	0.340
>=18	1	11.5	3	10.7	55	8	1423	243	0.92	0.48	1.76	0.00		
18–64	6	26.9	7	25.0	393,999	5481	1,637,038	21,011	0.91	0.85	0.97	72.11		
>=65	7	61.5	16	57.1	356,212	1519	802,583	3991	0.97	0.92	1.02	78.23		
Study design													0.11	0.740
Cohort-Retrospective	8	66.7	15	57.7	563,409	3,926	1,618,444	12,821	0.94	0.89	0.98	89.21		
Test-Negative	4	33.3	11	42.3	584	63	2563	270	0.96	0.85	1.08	0.00		
Outcome setting													1.78	0.410
Primary Care	2	16.7	3	11.5	247,990	1630	906,940	8145	0.88	0.72	1.08	80.10		
Hospital	8	66.7	18	69.2	310,162	885	765,760	2544	0.93	0.89	0.98	61.20		
Both	3	25.0	5	19.2	426,133	10,304	1,560,071	33,138	0.98	0.92	1.04	85.13		
Outcome determination													0.55	0.460
Laboratory confirmed	5	41.7	13	50.0	6753	63	111,727	270	0.98	0.87	1.09	0.00		
Clinical Codes	7	58.3	13	50.0	643,828	3,926	1,757,878	12,821	0.93	0.89	0.98	90.60		
Funding													0.36	0.550
Other	7	58.3	18	69.2	290,879	1122	728,565	2996	0.95	0.90	1.00	57.05		
Pharma	5	41.7	8	30.8	402,715	4223	1,398,836	15,213	0.92	0.85	1.00	92.69		
Risk of bias													2.05	0.150
Low	9	75.0	22	84.6	357,782	2433	987,002	7500	0.96	0.92	1.00	78.95		
Moderate	3	25	4	15.4	146,586	768	647,702	4910	0.87	0.77	0.99	65.09		

IRO, influenza related outcome. IV, Influeza vaccine. aRR, adjusted relative risk of IRO comparig cell-cultured with egg-based vaccine: <1 favors cell culture vaccine. CI, confidence interval. * Excluded results: Bruxvoort 2019b; Izurieta 2019a; Izurieta 2019c; Klein 2020b; Martin 2020a; Martin 2020b; Fu Tseng 2019a; Fu Tseng 2019c; Fu Tseng 2019d; Fu Tseng 2019e, Izurieta 2020b, and Izurieta 2020d. ^†^ When totals are higher that 12 is because one study reports more that one result in the same age category. ^¶^ Adjusted relaive vaccine effectiveness estimated as (1 − aRR) ∗ 100. ^§^ Statistic for assessing heterogeneity. It estimates the proportion of variation between the effect sizes due to heterogeneity relative to the pure sampling variation. I^2^ > 50 indicates substantial heterogeneity. ** The Q homogeneity test evaluates whether the effect sizes are the same across the results. We use the significance level ≥ 0.1. The test does not estimate the magnitude of the heterogeneity.

## Data Availability

The data presented in this study is available in the manuscript and Appendix A.

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
