# Peer review of "Relative Effectiveness of Cell-Cultured versus Egg-Based Seasonal Influenza Vaccines in Preventing Influenza-Related Outcomes in Subjects 18 Years Old or Older: A Systematic Review and Meta-Analysis"

_ijerph, 2022, doi:10.3390/ijerph19020818_

Round 1

Reviewer 1 Report

The manuscript entitled "Relative effectiveness of cell-cultured versus egg-based seasonal influenza vaccines in preventing influenza-related outcomes in subjects 18 years old or older: a systematic review and meta-analysis" is very interesting and well organized review article. In this review, the authors performed a systematic review and meta-analysis of the adjusted relative vaccine effectiveness (arVE) of seed cell-cultured influenza vaccines (ccIV) compared to egg-based influenza vaccines (eIV) in preventing laboratory-confirmed influenza related outcomes (IRO) in subjects 18 years old and older. Based on the meta-analysis, the authors concluded a low-to-moderate advantage of using ccIV over eIV in preventing IRO. Figures and Table are clear and well presented. References are adequate and updated. I consider this manuscript of interest but after moderate revision:

  1. The adaptive mutations in influenza H3N2 vaccine strains that affect vaccine effectiveness is a novel finding and is not limited also to egg-based manufacturing platform (DOI: 10.1007/s00705-016-2815-x). The adaptive consequences can also happen in cell-based vaccines (DOI: 10.1016/j.tim.2017.12.003). In the 12 studies that have been analyzed, the authors should correlate the impact of the cell-line used (VERO or MDCK) to the arVE or even discuss its possible impact in the discussion section.
  2. The analyzed studies are limited to the seasons 2017-2019. Retrospectively, is it possible that the authors compare the arVE in the analyzed data to those egg-based vaccines with known low effectiveness in the seasons 2012 and 2016.
  3. Line 19: replace “in subjects 18 years and over” with “in subjects≥18 years old.

Author Response

Thank you for your comments.

  1. The adaptive mutations in influenza H3N2 vaccine strains that affect vaccine effectiveness is a novel finding and are not limited also to egg-based manufacturing platforms (DOI: 10.1007/s00705-016-2815-x). The adaptive consequences can also happen in cell-based vaccines (DOI: 10.1016/j.tim.2017.12.003). In the 12 studies that have been analyzed, the authors should correlate the impact of the cell-line used (VERO or MDCK) to the arVE or even discuss its possible impact in the discussion section.

We have reviewed with interest and attention the references provided by the reviewer. The only seed-cell-derived and the cell-culture influenza vaccine used in the studies were grown in MDCK cell lines (https://www.cdc.gov/flu/prevent/cell-based.htm). So, we cannot discuss the differences between VERO and MDCK platforms.

Although adaptative mutations have been described in both seed-mammalian cell-derived and chicken embryonated egg-derived vaccines, the mutations considered relevant in terms of vaccine effectiveness are those that revert the virus to its original avian host (Skowronski, D. M., Janjua, N. Z., De Serres, G., Sabaiduc, S., Eshaghi, A., Dickinson, J. A., Fonseca, K., Winter, A.-L., Gubbay, J. B., Krajden, M., Petric, M., Charest, H., Bastien, N., Kwindt, T. L., Mahmud, S. M., Van Caeseele, P., & Li, Y. (2014). Low 2012-13 influenza vaccine effectiveness associated with mutation in the egg-adapted H3N2 vaccine strain not antigenic drift in circulating viruses. PloS One, 9(3), e92153. https://doi.org/10.1371/journal.pone.0092153).

Our results for the 2017-2018 season, with no A(H3N2) vaccine strain mismatch with the circulating A(H3N2), do not falsify this hypothesis.

2. The analyzed studies are limited to the seasons 2017-2019. Retrospectively, is it possible that the authors compare the arVE in the analyzed data to those egg-based vaccines with known low effectiveness in the seasons 2012 and 2016.

Our goal was expressly to review the effect of seed-cell derived vaccines to exclude the impact of seed-strain avian adaptation when the seed strain was obtained in eggs. Although we understand the interest in comparing vaccines with seed viruses obtained in different platforms, we consider that currently and in the future, the relevant evidence is comparing seed-cell derived with egg-based vaccines.

3. Line 19: replace “in subjects 18 years and over” with “in subjects≥18 years old.

We have replaced the expression as suggested.

Reviewer 2 Report

The authors report a meta-analysis of ccIV vs eIV vaccine effectiveness in several influenza seasons. The topic of the study is relevant and the data provided is clear, well-written and of interest to the audience. Before acceptance, authors should address the following comments:

Line 27) Should 0.1% be used instead of 01%?

Line 46-47) Authors should mention that some influenza seasons report vaccines effectiveness < 10%, specially for A/H3N2 viruses.

Line 149) Authors should mention how they do calculate the aRR in a more comprehensible manner. As it is, it is diffuse and might lead to misunderstanding

Line 240) Which are units of aRRs mentioned? Are these z-score values?

Line 349) Is the second goal related to the comparison of ccIV vs eIV including all influenza types?

Line 396-397) The explanation provided by the authors should be better discussed. If the authors do not encounter differences between ccIV vs eIV in the 2018-19 season, it is difficult to accept that this is because the group 1 component was egg-derived in all cases. Specifically, if the H3N2 component is produced in eggs and cells, depending on the vaccine, there should be significant differences in H3N2 protection in the case that the cell-derived H3N2 component of the vaccine performs better than egg-derived one. Authors should comment on that.

Line 410-411) Did the authors test the adequacy of the sample in each study to be statistically significant for inclusion in the meta-analysis?

General comment in the discussion section) There a number of reviews comparing ccIV vs eIV effectiveness. The manuscript would benefit from comparison to the state of the art.

Author Response

Thank you for your encouraging comments.

Line 27) Should 0.1% be used instead of 01%?

Thank you for pointing to this mistyping. The correct figure is -0.01%. 

We have amended the result in line 27. Now it reads: -0.01%

Line 46-47) Authors should mention that some influenza seasons report vaccines effectiveness < 10%, especially for A/H3N2 viruses.

This was meant to comment on the well-supported overall estimate of vaccine effectiveness with the actual vaccines.

We do not feel that here is opportune to describe the variability by strain, season, age of influenza vaccine effectiveness and provide a reference to an exhaustive review on the topic that supports the numbers we provide.

Line 149) Authors should mention how they do calculate the aRR in a more comprehensible manner. As it is, it is diffuse and might lead to misunderstanding.

We consider that in lines 145 to 148, there is a clear explanation of how adjusted relative risks and adjusted relative vaccine effectiveness was obtained. Note that we write:

The published results and their confidence intervals were incidence rates ratios, risk ratios, or adjusted odds ratios (aOR) depending on the study design. Under the rare-diseases assumption, we assumed these measures as unbiased estimators of the adjusted relative risk (aRR)[41] and arVE as (1-aRR)*100[42].

The two added references support our assumptions and course of action.

Line 240) Which are units of aRRs mentioned? Are these z-score values?

The aRR measures the adjusted incidence rates of IRO in those exposed to the cell-based vaccine compared to those exposed to the egg-based vaccine. We computed the results we report from the estimates in the publications and their confidence intervals. We declare de data for meta-analysis in STATA as the log odds-ratios, log relative risks or log hazard- ratios and the computed standard errors based on the specified 95% CI variables. We calculated the results in the text using a restricted maximum likelihood. So these are not z-scores but computed maximum likelihood functions.

Line 349) Is the second goal related to the comparison of ccIV vs eIV including all influenza types?

See lines 84-85, where we wrote that as secondary objectives, we looked to evaluate the arVE of ccIV, compared to eIV in preventing IRO with influenza.

In lines 87-89, we wrote: We defined IRO as any clinical outcome related to influenza, determined as IRO with laboratory-confirmed influenza or IRO with influenza-specific clinical codes.

Line 396-397) The explanation provided by the authors should be better discussed. If the authors do not encounter differences between ccIV vs eIV in the 2018-19 season, it is difficult to accept that this is because the group 1 component was egg-derived in all cases. Specifically, if the H3N2 component is produced in eggs and cells, depending on the vaccine, there should be significant differences in H3N2 protection in the case that the cell-derived H3N2 component of the vaccine performs better than egg-derived one. Authors should comment on that.

We agree that this point is to be clarified. We have added the following text "The results in the 2018-2019 (and the only study in the 2019-2020) do reinforce the observation in the 2017-2018 season of a favourable impact, compared to the egg-based vaccine, of the seed-cell-based vaccine. When there is no mismatch between the vaccine and the circulating strains, egg adaptation has a negative impact. In the presence of a mismatch, it is the mismatch the main driver to low vaccine effectiveness." Now, lines 400-405.

Line 410-411) Did the authors test the adequacy of the sample in each study to be statistically significant for inclusion in the meta-analysis?

The sample size in itself was not a criterion for exclusion. If we had excluded those small "imprecise" studies, we would have incurred small-studies exclusion bias (we explore and describe this in the publication bias section) and provided biased results.

General comment in the discussion section) There are a number of reviews comparing ccIV vs eIV effectiveness. The manuscript would benefit from comparison to the state of the art.

As we are aware, the relevant review on the topic is the one published in 2020 by ECDC (ref 27 in the text). The results and conclusion of this ECDC review justify our evaluation of the evidence published after the ECDC author's closed their search.

Reviewer 3 Report

This is an interesting review in which the authors describe the relative effectiveness of cell-cultured versus egg-based seasonal influenza vaccines in preventing influenza-related outcomes . The review is well written, should be published in this journal.   

However, there is one point that I would like the author to confirm.

page 6, line 205.

the U.S. Seven publications reported 14 results in the 2017-2028 influenza season,

Is the year in this section? 

Author Response

Thank you for your encouraging comments.

page 6, line 205.

the U.S. Seven publications reported 14 results in the 2017-2028 influenza season,

Is the year in this section? 

Thank you for noticing the mistyping. We have now amended it in the revised version of the manuscript. 

We have amended it, now it reads, "the U.S. Seven publications reported 14 results in the 2017-2018 influenza season,"

Round 2

Reviewer 1 Report

No further comments